# Evidence that DNA polymerase δ contributes to initiating leading strand DNA replication in *Saccharomyces cerevisiae*

Marta A. Garbacz[1], Scott A. Lujan [1], Adam B. Burkholder[2], Phillip B. Cox[1], Qiuqin Wu[3], Zhi-Xiong Zhou[1], James E. Haber[3] & Thomas A. Kunkel[1]

To investigate nuclear DNA replication enzymology in vivo, we have studied *Saccharomyces cerevisiae* strains containing a *pol2-16* mutation that inactivates the catalytic activities of DNA polymerase ε (Pol ε). Although *pol2-16* mutants survive, they present very tiny spore colonies, increased doubling time, larger than normal cells, aberrant nuclei, and rapid acquisition of suppressor mutations. These phenotypes reveal a severe growth defect that is distinct from that of strains that lack only Pol ε proofreading (*pol2-4*), consistent with the idea that Pol ε is the major leading-strand polymerase used for unstressed DNA replication. Ribonucleotides are incorporated into the *pol2-16* genome in patterns consistent with leading-strand replication by Pol δ when Pol ε is absent. More importantly, ribonucleotide distributions at replication origins suggest that in strains encoding all three replicases, Pol δ contributes to initiation of leading-strand replication. We describe two possible models.

[1] Genome Integrity and Structural Biology Laboratory, National Institute of Environmental Health Sciences, NIH, DHHS, Research Triangle Park, Durham, NC 27709, USA. [2] Integrative Bioinformatics Support Group, National Institute of Environmental Health Sciences, NIH, DHHS, Research Triangle Park, Durham, NC 27709, USA. [3] Department of Biology, Rosenstiel Basic Medical Sciences Research Center, Brandeis University, Waltham, MA 02454, USA. Correspondence and requests for materials should be addressed to T.A.K. (email: kunkel@niehs.nih.gov)

Replication of the undamaged eukaryotic nuclear genome is largely conducted by three members of the B family of DNA polymerases. DNA polymerase α-primase synthesizes short RNA–DNA primers to initiate replication and DNA polymerases δ and ε (Pols δ and ε) then perform the bulk of DNA chain elongation[1–5]. Because these polymerases only synthesize DNA in the 5′-to-3′ direction, the leading strand of duplex DNA is normally replicated in a largely continuous fashion, while the lagging strand is synthesized discontinuously as Okazaki fragments. Considerable evidence now suggests that in the absence of stress, lagging-strand replication is primarily conducted by Pol δ and leading-strand replication is primarily conducted by Pol ε (reviewed in ref. [4]). For example, a variant of Saccharomyces cerevisiae Pol ε produces specific mismatches[6] and incorporates an increased number of ribonucleoside tri-phosphates (rNTPs)[7] during DNA synthesis in vitro, and these same behaviors are seen in vivo during leading-strand DNA replication in S. cerevisiae and Schizosaccharomyces pombe[8–13]. The analogous situation is true for variants of budding and fission yeast Pols α and δ, whose strand-specific incorporation of both mismatches and ribonucleotides are consistent with lagging-strand replication[10,11,14,15]. This model is strongly supported by elegant studies of DNA replication catalyzed by yeast replication forks in vitro[1–3]. Moreover, it does not exclude that Pol δ is also involved in a smaller percentage of leading-strand replication[4]. For example, elegant studies in S. pombe by Carr and colleagues suggest that Pol δ synthesizes about two kilobases of DNA on both DNA strands upon replication restart after pausing at the RTS1 locus[16]. They also reported a slight excess of ribonucleotides incorporated into the leading strand during replication by Pol δ [13], and suggested that it may occasionally be recruited to initiate leading-strand replication. More recently, Diffley and colleagues have described a pulse-chase experiment in vitro of replication products made during replication with purified yeast proteins[17], and they too have suggested that Pol δ participates in initiating leading-strand replication.

On the other hand, studies performed in the 1990s revealed that SV40 origin-dependent DNA replication in vitro requires Pols α and δ [18–20], but not Pol ε. Moreover, S. cerevisiae pol2-16[21–23] and S. pombe cdc20[ΔN-term][24] mutant strains are viable despite having in-frame deletions of Pol ε polymerase and exonuclease domains while leaving intact the C-terminal domain that controls cellular responses to DNA damage. These facts are consistent with an alternative model that is supported by a recent study[25] proposing that Pol δ is the major replicase for both strands and that Pol ε simply proofreads errors made by Pol δ as it replicates the leading DNA strand.

Here we provide evidence to distinguish between these two models and to inform our understanding of the initiation of nuclear DNA replication. We begin by examining the phenotypes of S. cerevisiae pol2-16 mutants in two strain backgrounds, and compare their phenotypes to those of pol2-4 mutants that lack exonuclease activity but retain polymerase activity[26]. These comparisons support the idea that Pol ε is the main DNA polymerase used to synthesize most of the undamaged leading strand during nuclear DNA replication in yeast. We then examine ribonucleotide incorporation into DNA in the pol2-16 mutant and conclude that in the absence of Pol ε polymerization activity, Pol δ replicates both DNA strands, but does so poorly and with severe consequences on genome stability. Finally, we present ribonucleotide incorporation data with variants of Pols α, δ, and ε that strongly support the hypothesis that Pol δ contributes to initiation of leading-strand replication in yeast by synthesizing DNA of both strands at replication origins.

## Results

### In vivo phenotypes of pol2-16 mutants.
Seminal studies by Wittenberg and colleagues[21] and by Campbell and colleagues[22] demonstrated that S. cerevisiae strains containing an in-frame deletion of residues 176–1134 of the Pol ε catalytic subunit (pol2-16; Fig. 1a) are viable. Because this deletion removes the DNA polymerase and 3′-to-5′ exonuclease activities of Pol ε but leaves C-terminal residues intact, the survival of these pol2-16 strains demonstrates that yeast cells can replicate their nuclear genome in the absence of these two catalytic activities of Pol ε. In order to better understand the consequences of loss of Pol ε catalytic activities on replication in yeast cells, we constructed the pol2-16 mutation and examined its properties. Because different genetic backgrounds can affect phenotype, and because this has been offered as one explanation for differences between our studies and that of Johnson et al.[25], we constructed the pol2-16 mutant in two strain backgrounds. One is the yeast strain that we have previously used to investigate the role of Pol ε in nuclear DNA replication, designated Δ7. The other is the commonly used strain background, W303. To distinguish between the two replication models, we compared pol2-16 phenotypes in both backgrounds to those of analogous wild-type yeast and to strains lacking Pol ε's 3′-exonuclease activity (pol2-4) due to substitution of alanine for two negatively charged residues (D290 and E292) that are essential for proofreading of mismatches and ribonucleotides[26,27]. We anticipated that a comparison of the phenotypes of the pol2-16 and pol2-4 mutants would be informative regarding loss of the one or both of the catalytic activities of Pol ε on survival, growth characteristics, and the role of Pol ε in normal DNA replication.

We constructed heterozygous diploid yeast (pol2-16/POL2 and pol2-4/POL2) by replacing one of the POL2 alleles and explored the phenotypes of meiotic progeny. On the third day after tetrad dissection, wild-type and pol2-4 yeast formed haploid spore colonies that are visible by eye at both 23 and 30 °C (Fig. 1b, c). At the same time, pol2-16 spore colonies are not visible to the naked eye, although micro-colonies are observable (Fig. 1d). After incubation for 12 days, some, but not all, of the pol2-16 mutants formed small spore colonies. In the Δ7 background, pol2-16 spore viability was 64% at 23 °C and 19% at 30 °C; in the W303 background, the pol2-16 spore viability was 77% at 23 °C and 44% at 30 °C. The differences in pol2-16 spore viability between 23 °C and 30 °C were statistically significant (Supplementary Fig. 1b). The viability of wild-type spores was not affected by temperature −93% at both 23 and 30 °C in the Δ7 background and 94% at 23 °C and 93% at 30 °C in the W303 background (Supplementary Fig. 1c). Freshly isolated pol2-16 spore colonies have doubling times that are strongly increased as compared to the pol2-4 mutant and wild-type yeast (Fig. 1e).

Western blot analysis of Pol2p levels revealed that the truncated Pol2p in the pol2-16 yeast is unstable, which manifest as multiple bands of degradation products (Fig. 1f and Supplementary Figs. 5 and 6). The level of non-degraded Pol2p in pol2-16 yeast is about 70% lower than wild type (Fig. 1g).

Consistent with earlier studies[21–23], our pol2-16 mutants have larger cell sizes than wild type (Fig. 2a), and they progress more slowly through the cell cycle (Fig. 2b). Moreover, the DNA content of pol2-16 mutant cells is higher than that of wild-type or pol2-4 mutant cells (Supplementary Fig. 1d), and this DNA is aberrantly distributed (Fig. 2a and Supplementary Fig. 1e). When pol2-16 spore colonies were resuspended and plated onto complete medium, colonies formed that ranged from barely macroscopic to close to wild-type in size (Fig. 2c). This indicates rapid accumulation of suppressors that improve pol2-16 fitness (parental suppressors are precluded; Supplementary Tables 1 and 2). To minimize the effects of such suppressors, subsequent

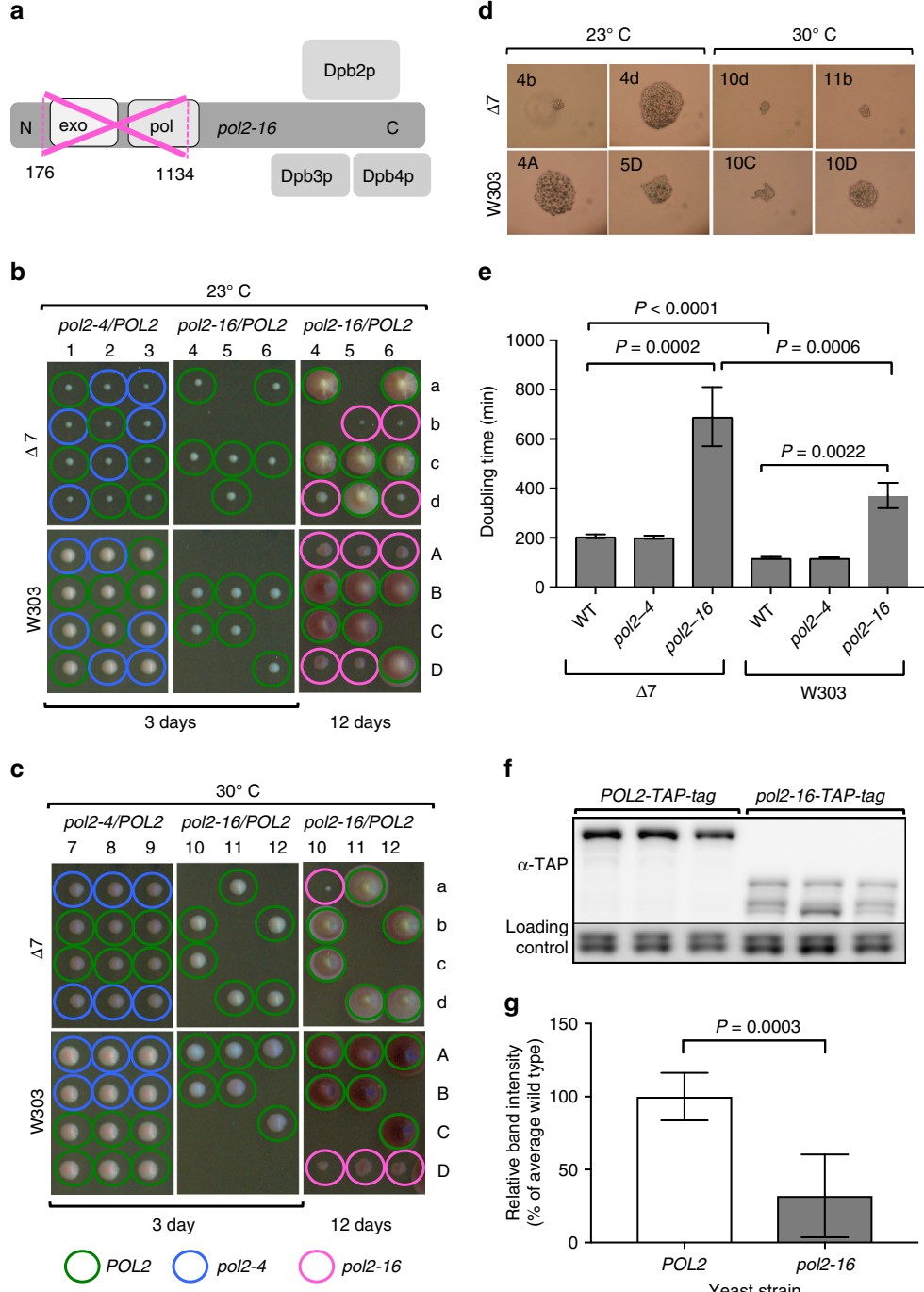

**Fig. 1** Pol ε catalytic domains are critical for yeast growth. **a** Schematic representation of DNA Polymerase ε (Pol ε). The *S. cerevisiae* holoenzyme consists of the catalytic subunit (Pol2p) and three auxiliary subunits: Dpb2p, Dpb3p, and Dpb4p[52–55]. Cryo-electron microscopy has shown that Pol2p has two lobes tethered by a flexible linker[43]. Active polymerase and exonuclease domains are in the N-terminal lobe. The *pol2-16* mutant has an in-frame deletion of the fragment of catalytically active lobe (amino acids 176–1134). **b**, **c** Tetrad analysis of *pol2-16/POL2* and *pol2-4/POL2* heterozygous diploids in two yeast backgrounds, Δ7, and W303, at 23 °C **b** and 30 °C **c**. 1–12 are dissected tetrads, A–D, and a–d are haploid spore colonies. Images were taken after 3 and 12 days. Genotypes were confirmed via PCR (*pol2-16*, red circles) or sequencing (*pol2-4*, blue). Wild-type colonies are circled in green. The lack of both Pol ε catalytic domains (*pol2-16*) causes severe growth defects. Exonuclease inactivation alone (*pol2-4*) does not. **d** Microscopic images of *pol2-16* colonies taken 3 days after tetrad dissections. **e** Doubling times of *pol2-16* and *pol2-4* mutants compared to wild-type yeast. Doubling times were estimated from optical density at 600 nm of cultures grown at 23 °C. Error bars represent standard deviations (n = 4–6 yeast cultures, two or three from two independent isolates). Unpaired two-tailed *t* tests with Welch's correction yielded *p* values (*P*). The doubling time of the *pol2-16* mutant is about threefold longer than of the wild-type and *pol2-4* yeast in both Δ7 and W303 backgrounds. The difference in the doubling times between the wild-type Δ7 and W303 backgrounds may be due to one or more of over 10,000 SNPs detected by the whole-genome sequencing. **f** Western blot detection of Pol2p level in whole-cell extracts. Presented are bands for three independent isolates of strains bearing *POL2* or *pol2-16* in fusion with TAP-tag. Immunoblotting was performed using an antibody to TAP-tag or PSTAIR (loading control). **g** Relative band intensity. Error bars represent standard deviations (n = 6–7 independent yeast isolates). Unpaired two-tailed *t* tests with Welch's correction yielded *p* values (*P*)

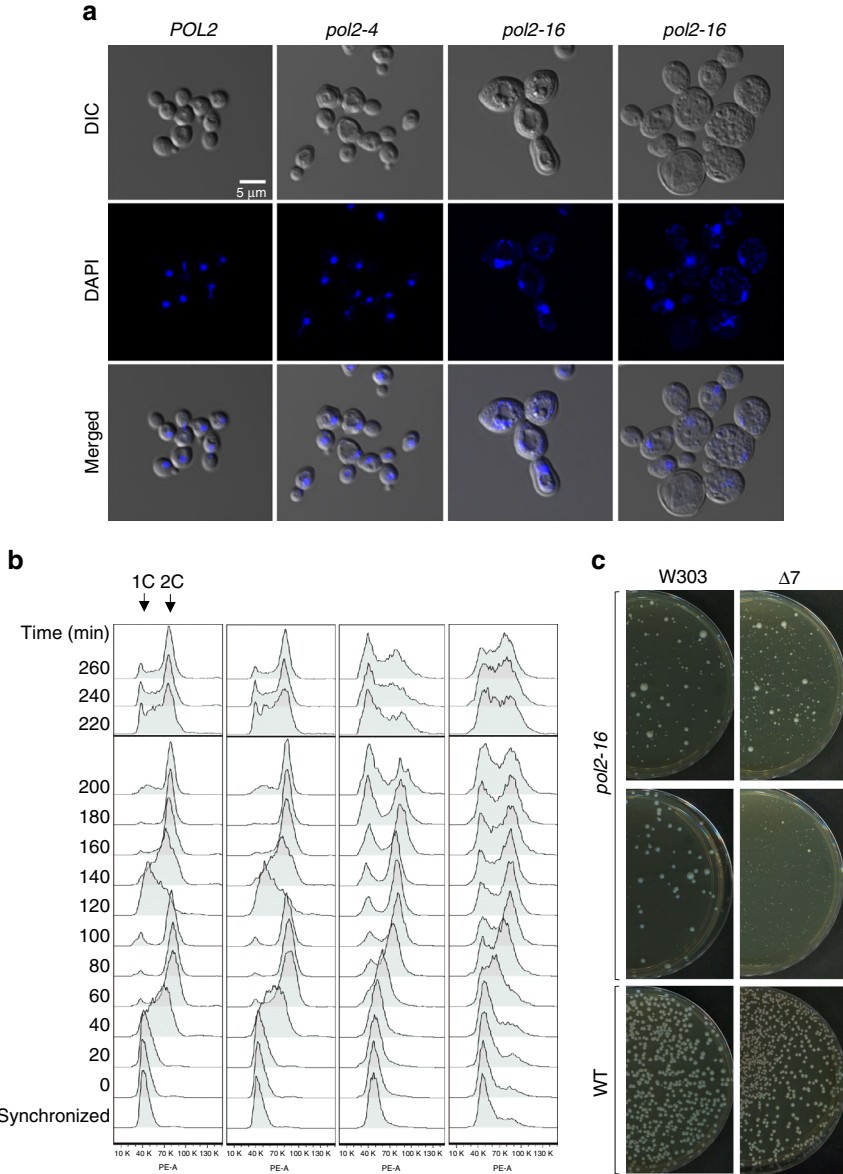

**Fig. 2** Phenotypes of wild-type, *pol2-4*, and *pol2-16* mutants. **a** Confocal microscope images of DAPI stained, exponentially growing yeast in the synthetic complete media, fixed with 70% ethanol. The *pol2-16* cells are much bigger than wild-type and *pol2-4* cells and have aberrantly distributed DNA. **b** Flowcytometric analysis of cell cycle progression. Yeast cells were α-factor arrested in G1 phase, then released into complete media, sampled every 20 min, fixed with 70% ethanol, stained with propidium iodide, and then analyzed with flow cytometry (Methods). 1C and 2C indicate the DNA contents. **c** Colony size heterogeneity in the *pol2-16* outgrowths. Resuspended *pol2-16* and *POL2* spore colonies, were plated on solid complete media and incubated at 23 °C for 6 and 4 days, respectively

experiments (with one exception, see below) were performed with *pol2-16* spore colonies freshly isolated from progeny of heterozygous *pol2-16/POL2* diploids.

**Replication enzymology in the W303 background**. We examined the genome-wide density of ribonucleotides incorporated by variants of the three major replicases, Pol α (*pol1-L868M*), Pol δ (*pol3-L612M*), and Pol ε (*pol2-M644G*) in the W303 strain background. Relative to their wild-type parents, these polymerases all have an elevated ability to incorporate ribonucleotides into DNA[7,27]. This property can be visualized when ribonucleotide excision repair (RER), the primary mechanism of ribonucleotide removal, is absent (i.e., in the *rnh201Δ* background; reviewed in ref. [10]). This in vivo analysis uses HydEn-seq[11], a

procedure that maps the locations of 5′ DNA ends via paired-end sequencing of genomic DNA after alkaline hydrolysis. In *rnh201Δ* strains, these positions largely indicate the locations of ribonucleotides incorporated by each DNA polymerase during replication, in a strand-specific manner. We therefore performed a meta-analysis of HydEn-seq data to examine ribonucleotide incorporation around 214 well-characterized replication origins in the W303 strain background[28]. The DNA ends in the *pol1-L686M rnh201Δ* and *pol3-612M rnh201Δ* strains were preferentially found in the newly synthesized lagging strand (Fig. 3a, c and Supplementary Data 1 and 2). In contrast, in *pol2-M644G rnh201Δ*, DNA ends were primarily present in the nascent-leading strand (Fig. 3b and Supplementary Data 1 and 2).

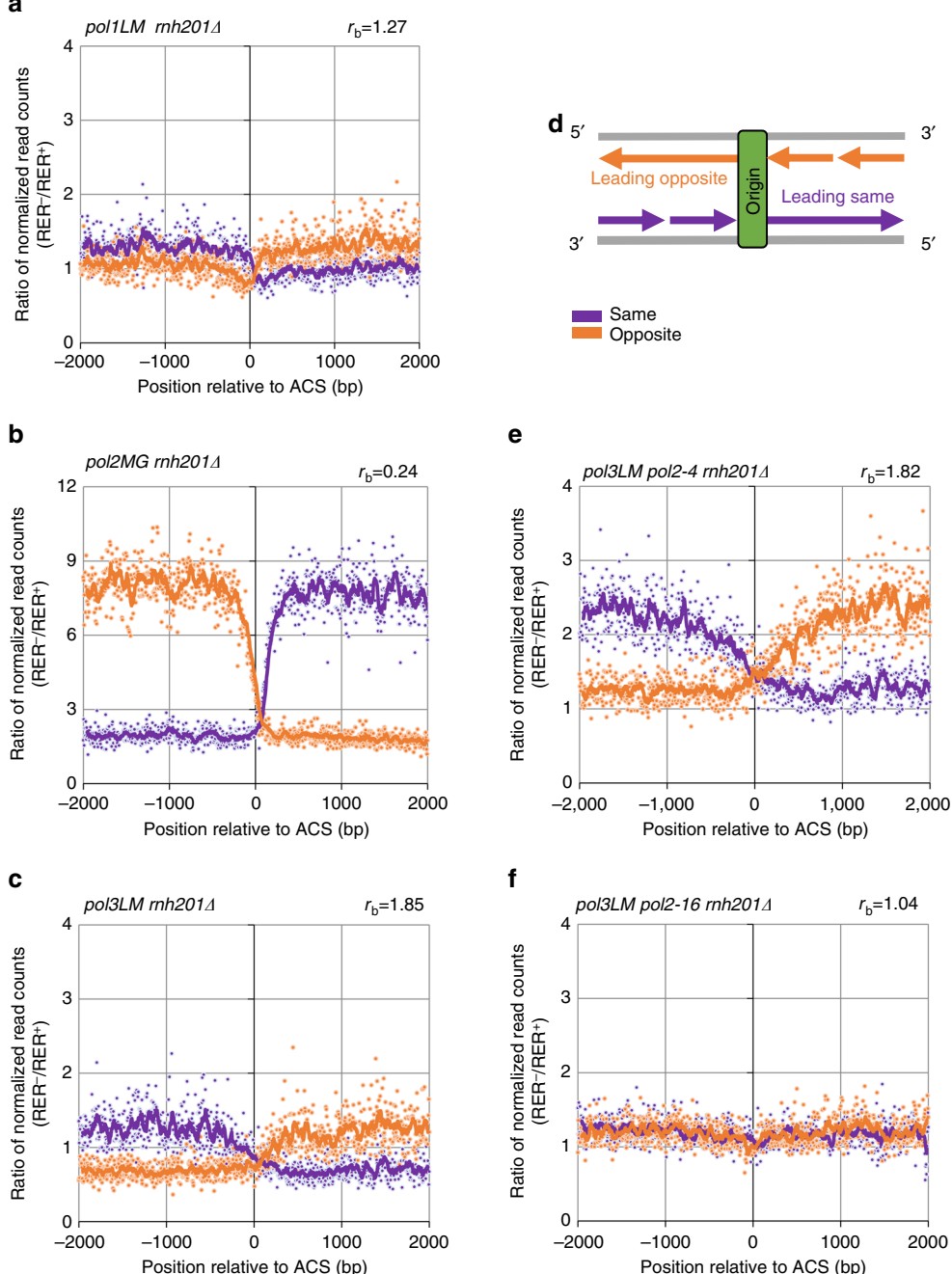

**Fig. 3** Ribonucleotides as biomarkers of replicase activity. Meta-analysis of ribonucleotides abundance in the vicinity of 214 replication origins analyzed in previous study[28], in bins of 5 bp, with 10-bin moving average trend lines. Read counts were normalized as described in Supplementary Methods. Shown are ratios of RER-deficient (RER$^-$) and RER-proficient (RER$^+$) strains bearing **a** *pol1L868M*, **b** *pol2M644G*, **c** *pol3L612M*, **e** *pol3LM pol2-4*, and **f** *pol3LM pol2-16* mutations. Values on the right above each chart are average lagging-over-leading strand biases ($r_b$) (see calculations in Supplementary Methods). For each panel, a representative of two independent measurements is presented. **d** Schematic representation of DNA replication in the vicinity of replication origin, depicting the leading and lagging strands orientations

**Pol δ synthesizes both strands in the *pol2-16* mutant**. In *pol2-16* mutants lacking Pol ε polymerase activity, Pol δ is a logical candidate for replicating leading-strand DNA. To test this possibility, we next used HydEn-seq to compare strand-specific genomic ribonucleotide abundance in *pol3LM rnh201Δ* strains containing either wild-type Pol ε, the *pol2-4* mutation, or the *pol2-16* mutation. As mentioned above, in the W303 double-mutant *pol3L612M rnh201Δ* strain, significantly more ribonucleotides are observed in the lagging strand (Fig. 3c), with a lagging-over-leading strand bias of 1.85 (calculations in

Supplementary Methods). A similar result was observed for the *pol3L612M rnh201Δ pol2-4* strain (Fig. 3e). These results are consistent with Pol δ being the major lagging-strand replicase and demonstrate that the leading-strand biases observed in the *pol2-M644G rnh201Δ* mutant are due to Pol ε polymerase activity rather than to defective Pol ε proofreading of Pol δ errors on the leading strand. However, in the triple mutant *pol3L612M rnh201Δ pol2-16* strain, the strand bias is greatly reduced (1.04; Fig. 3f). These results suggest that Pol δ replicates both nascent strands in the absence of Pol ε catalytic activity, albeit with

reduced efficiently that manifests as slow growth phenotypes for *pol2-16* mutants (Figs. 1 and 2).

**Pol32 and Pif1 are required in the *pol2-16* mutant**. When Pol ε is inactivated, kilobases of break-induced replication (BIR) synthesis are accomplished in a Pol δ-dependent manner[29]. While the Pol32 subunit of Pol δ and the 5′–3′ Pif1 helicase are not required for normal DNA replication, they are required for BIR, where recombination establishes a non-canonical replication fork[29–31]. To see whether Pol32 and Pif1 are required for extensive leading-strand DNA replication by Pol δ in the *pol2-16* mutant, we performed the series of crosses outlined in the Supplementary Fig. 3d. In this series of crosses, during propagation of the initial ASY102 *pol2-16* strain, as well as of haploid strains obtained from tetrad dissections, unknown suppressor(s) of the *pol2-16* slow growth phenotype could have been acquired. The distribution of spore phenotypes from a subsequent cross (Supplementary Figs. 3a and 4) is in agreement with a single suppressor that segregates independently of *pol2-16* in this diploid (Supplementary Tables 3 and 4). Two different fast-growing *pol2-16* isolates from progeny of heterozygous *pol2-16/POL2* diploids were crossed with *pol32Δ* or with *pif1Δ* isolates from the same series of crosses (Supplementary Fig. 3b, c). No *pol2-16 pol32Δ* meiotic segregants arising from a cross were viable. In contrast, half of the *pol2-16* segregants, presumably those with the suppressor, grew well (Supplementary Fig. 3). The *pol2-16 pif1Δ* double-mutant segregants did grow, but grew poorly.

**Pol δ participates in leading-strand replication at origins**. We observed that ribonucleotide density at origins is elevated in both nascent strands in the strains bearing Pol δ variants and minimal in the *pol2-M644G rnh201Δ* strains, in both the W303 (Supplementary Fig. 2 and Supplementary Data 2) and Δ7 strain backgrounds[11]. Based on these observations, we decided to further explore the replication enzymology of the leading strand in vivo to test the possible involvement of Pol δ in leading-strand initiation. After subtracting the HydEn-seq end densities of the RER-proficient strains from their RER-deficient analogs (Supplementary Methods, Supplementary Fig. 2, and Supplementary Data 2), we solved a system of simultaneous equations to account for ribonucleotide incorporation by each variant polymerase during replication across origins (Supplementary Methods). This meta-analysis used 214 well-behaved replication origins, centered on the 5′-end of the autonomously replicating sequence (ARS) consensus sequence (ACS). The results (Fig. 4) indicate that the fraction of DNA synthesis conducted by each variant polymerase differs immediately proximal to the point of initiation. As an example, for the most promiscuous DNA polymerase for ribonucleotide incorporation, Pol α[7], the total amount of DNA synthesis at the origin (Fig. 4a, in red) is low because the DNA primers it synthesizes are only about 10–20 nucleotides long. When the results with Pol α are used to estimate where Pol α begins to synthesize DNA at an average replication origin, the results reveal a peak of DNA synthesis centered just upstream of the ACS (Fig. 4b). On the other hand, Pol ε synthesizes much more DNA during replication and this synthesis is maximal by about 300 nucleotides downstream of the ACS (blue line in Fig. 4a). This is consistent with its role as the major leading-strand replicase. However, the beginning of this synthesis peaks about 180 nucleotides downstream of the ACS (Fig. 4a, b), approximately one Okazaki fragment length after the peak for initiating short primers synthesized by Pol α.

At locations >400 nucleotides upstream of the ACS (left of the ACS in Fig. 4a, b), the ribonucleotide incorporation data indicate that total synthesis by Pol δ (green in Fig. 4a) is maximal and

almost as high as the maximum for Pol ε seen on the leading strand. This pattern is anticipated by the model wherein Pol δ primarily replicates the lagging strand while incorporating ribonucleotides at the lowest rate among the three replicases[7]. More interestingly, after subtracting canonical lagging-strand synthesis (Supplementary Methods, Eq. 20, and Supplementary Data 3), a peak of Pol δ synthesis remains, with the beginning of this tract peaking about 10 nucleotides downstream of the Pol α initiation peak, well before the beginning of synthesis by Pol ε. Moreover, the standard deviation for the Pol δ tract length is much smaller than the track length ($0-12 \ll 140-180$), indicating that Pol α-to-Pol ε transfers (Pol δ tract = 0) are exceedingly rare. These results suggest that Pol δ synthesizes both DNA strands over a short distance at most replication origins, after which synthesis of the leading strand is primarily conducted by Pol ε.

**Discussion**

Studies by Kesti et al.[21] and by Dua et al.[22] showed that *pol2-16* is able to complement the lack of growth at increased temperature of yeast strains bearing temperature-sensitive alleles of *pol2*. Based on that analysis, they concluded that the Pol ε catalytic domains are dispensable for DNA replication and cell viability. However, the time between haploid strain construction and commencement of complementation experiments risks acquisition of suppressor mutations that could mask the full effects of *pol2-16*. We observed an apparent rapid acquisition of suppressors in spore colonies, and thus decided to minimize the number of generations during which yeast cells experience selective pressure against the effects of *pol2-16*. We constructed *pol2-16/POL2* heterozygous diploids from which we isolated haploid *pol2-16* meiotic progeny to measure phenotypes while minimizing accumulation of suppressors. Kesti et al.[21] also created *pol2-16/POL2* heterozygous diploids and observed that *pol2-16* spores "often germinated but grew into colonies only infrequently (~10% of expected segregants)." Kesti et al.[21] suggested that these phenotypes reflect the importance of the N-terminal catalytic activities of Pol ε during the first few cell cycles. Given the rapid suppressor accumulation that we observe here, we do not disagree with this interpretation. However, we further suggest that these phenotypes reflect the importance of Pol ε for replicating leading-strand DNA in all cell divisions, with the chance of observing the phenotypes of the *pol2-16* mutation being highest before suppressors come to dominate the population. In the future, it will be interesting to understand the nature of the suppression that is occurring in *pol2-16* cells and whether it is due to acquisition of mutations, epigenetic changes, or non-encoded metabolic adaptations that improve fitness.

The decreased level of Pol2p in the *pol2-16* freshly isolated spore colonies suggests that the truncated Pol2p in *pol2-16* is unstable. This suggests that the N-terminal portion of Pol2p (bearing the catalytic domains) is critical to stabilize the C-terminal part of Pol2p that is essential for yeast viability. It may further suggest that the sickness/near-inviability of *pol2-16* yeast could be due to replication initiation defects. The C-terminal part of Pol2p interacts with Dpb2p, which bridges the interaction of Pol ε with the GINS complex[32–34]. Pol ε and GINS are components of the CMGE helicase polymerase whose formation is essential for initiation of chromosomal DNA replication[32,33,35].

Overall, our studies of the *pol2-16* variant of Pol ε performed in two yeast strain backgrounds imply that the complete loss of Pol ε polymerase and exonuclease domains yields tiny spore colonies, rapid suppressor accumulation, greatly increased doubling time, increased cell size and aberrant DNA distribution. These phenotypes are not observed in the strain bearing the *pol2-4* mutation that lacks only Pol ε's proofreading activity. These data are

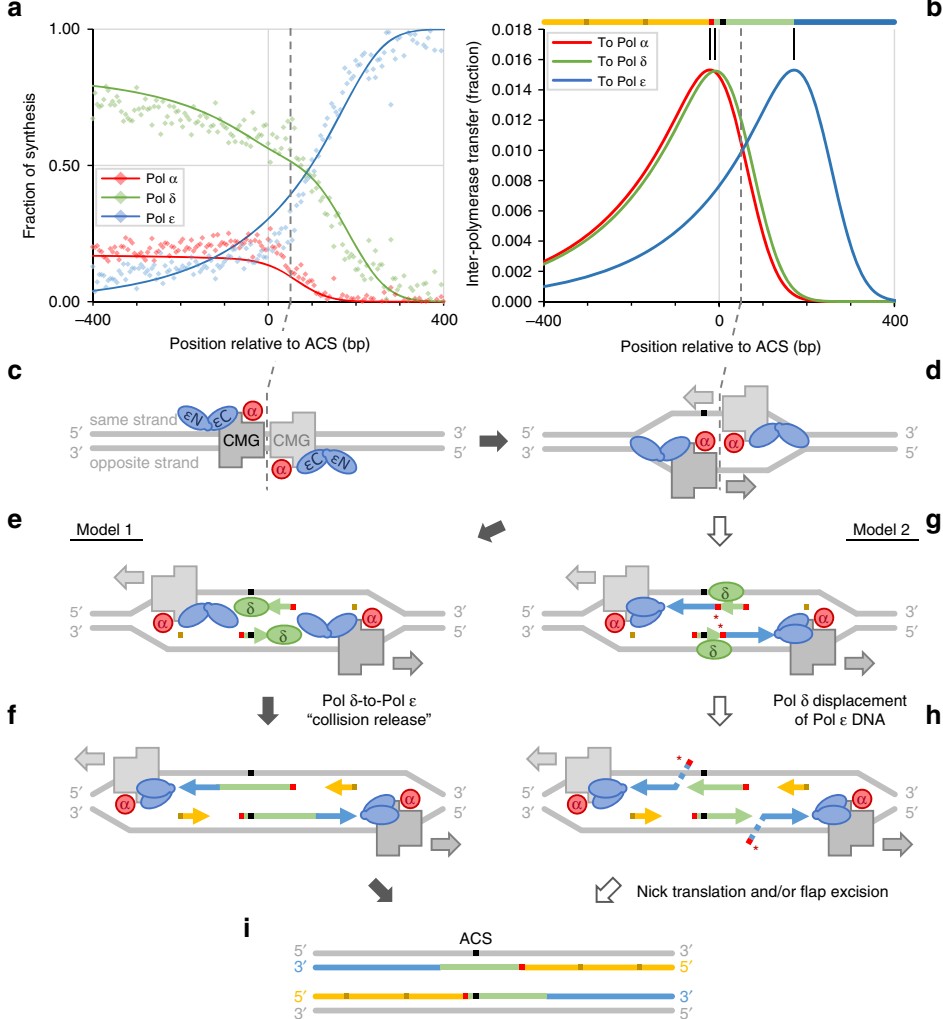

**Fig. 4** HydEn-seq-derived evidence for DNA polymerase δ participation in leading-strand synthesis at yeast origins. Red, green, and blue denote polymerases α, δ, and ε, respectively, or DNA tracts synthesized by same. Canonical Okazaki fragments (yellow), synthesized by Pols α (dark), and δ (light) are approximately positioned exemplars. *S. cerevisiae* origins (identified by Smith et al.[28]) are oriented such that ARS consensus sequences (ACS) are 5′–3′ beginning at position 0. **a** Diamonds represent the fraction of DNA strand synthesized by DNA Pols α, δ, and ε (5 bp bins; calculated from rescaled and background subtracted HydEn-seq end densities in *pol1-L868M rnh201Δ*, *pol3-L612M rnh201Δ*, and *pol2-M644G rnh201Δ* strains; see Supplementary Methods). Data for both strands were averaged (opposite strand reflected around +45 bp, the axis of strand symmetry; gray dashed line). Solid curves are regression models. **b** The fraction of inter-polymerase transfer events outside of canonical Okazaki fragment synthesis (extracted from regression models). The mode of each curve (vertical black line) suggests the most frequent synthesis tract (colored bars above). **c–i** Schematics of two non-exclusive models of polymerase action at yeast replication origins. DNA strands (colored bars) have the same horizontal scale as in **a** and **b**; polymerases (ellipses) and CMG helicases (gray polygons) are exaggerated; other components are omitted. εN and εC indicate N-terminal catalytic and C-terminal CMG-binding Pol ε domains. **c** Head-to-head dsDNA-binding CMG helicases. **d** Helicases transition to ssDNA-binding and translocate past one another, N-termini facing directions of travel (gray arrows). **e, f** Model 1. **e** Pol α associated with each replisome primes the leading strand that will be synthesized by the other (0.1% probability per bp translocated, from regression). Pol δ extends each nascent-leading strand. **f** Pols δ collide with respective replisomes, releasing 3′-termini to Pols ε. They assuming synthesis conformation to extend the leading strand. **g, h** Model 2. **g** Unidentified Pols α prime leading-strand replication (*; extended by Pols ε). The Pol α associated with each replisome primes the first Okazaki fragment (extended by Pol δ; destined to be longer than average). **h** Pol δ displaces the 5′ primer terminus of the nascent-leading strand, allowing nick translation or flap excision. **i** Synthesis patterns from both pathways indicate apparent Pol δ synthesis of both nascent strands at the origin

consistent with the hypothesis that Pol ε's polymerase activity, but not its proofreading activity, is crucial for efficient replication of undamaged chromosomal DNA that results in normal cell growth. These observations in vivo are in agreement with the recent in vitro data showing that full-length Pol ε is required for the maximal rate of leading-strand DNA replication[17].

The idea that Pol ε is the major leading-strand replicase has been questioned by Johnson et al.[25], who have suggested that differences in *S. cerevisiae* strain backgrounds could be misleading and that Pol δ is instead the major replicase for both the leading

and lagging strands. We do not consider this explanation to be likely given results in *S. cerevisiae* and *S. pombe* as well as mutation asymmetry around human origins in POLE-exo⁻ cancers, all of which support the idea that Pol ε is the major leading-strand replicase[6,9,11,13–15,36–38]. Nevertheless, because our previous studies were performed in the Δ7 strain background, we now include the W303 background when testing the two models. Our new ribonucleotide-incorporation data from the W303 background are remarkably similar to results in the Δ7 strain background[11], and are consistent with Pol α and Pol δ as primary

lagging-strand replicases and Pol ε as the primary leading-strand replicase.

Yeast are viable without Pol ε catalytic domains (*pol2-16*) but inviable with point mutations that abrogate Pol ε polymerase activity (*pol2*-D875A D877A)[22]. This suggests that at least one other polymerase can replicate the leading strand unless physically excluded from the primer terminus by inactive Pol ε. In the presence of PCNA, Pol δ can processively synthesize at least five kilobases of RPA-coated single-stranded DNA[39], which makes it the most probable polymerase to substitute for Pol ε in synthesis of the leading strand. Indeed, when replication fork progression is impeded at the *RTS1* locus in *S. pombe*, replication restarts through initial synthesis of both DNA strands by Pol δ[16]. The strand-specific ribonucleotide density in the *pol3LM rnh201Δ pol2-16* strain revealed no strand bias when the Pol ε catalytic domains are missing (Fig. 3), unlike the high bias in the *pol3LM rnh201Δ* strain, thereby suggesting that Pol δ replicates both the leading and lagging strands across entire genome when Pol ε is not present. Bulk leading-strand replication by Pol δ is very inefficient, which manifests as elongated S-phase and doubling time of yeast bearing the *pol2-16* mutation, likely due to a reduced rate of DNA unwinding by CMG that requires catalytic domain of Pol ε for the maxim unwinding rate[17]. Additionally, analysis of the synthetic genetic interactions of *pol2-16* with *pif1Δ* and *pol32Δ* suggest that for Pol δ to carry out extensive leading-strand synthesis, cells become reliant on normally non-essential replication factors such as Pif1 or Pol32. It is possible that the requirement for *POL32* and *PIF1* when Pol ε catalytic activity is absent is similar to their requirement in extensive DNA synthesis during BIR, where leading and lagging-strand DNA synthesis are not coupled[31] and where the initial DNA synthesis appears to be dependent on Pol δ, with Pol ε only being required at a later stage[29,40]. Further work will be needed to determine if the poor growth of the *pol2-16 pif1Δ* double mutant is partially due to loss of mitochondrial DNA due to *PIF1* deletion[41].

The idea that Pol ε is the major polymerase for leading-strand replication of undamaged DNA does not exclude Pol α or Pol δ participation in some fraction of leading-strand replication. Current mutagenesis and ribonucleotide incorporation data and eSPAN analyses[42] set the upper bound for such participation in yeast at between 2 and 23%, with the preponderance of evidence supporting the lower end of that range[4]. Using ribonucleotide bin sizes of 300 bases, Carr and colleagues reported that *S. pombe* Pol δ may occasionally initiate leading-strand replication[13]. More recently, a pulse-chase experiment of DNA synthesis by a minimal yeast replisome reconstituted using purified proteins and a single replication origin[17] led to the same interpretation. Our high-resolution ribonucleitide incorporation data in vivo (5-bp bins; Fig. 4b) are consistent with these two studies, and strongly imply that Pol δ synthesizes both nascent strands at most replication origins over a distance of about 160 nucleotides. Elegant biochemical data that include cryo-electron microscopy structures of the minimal replication fork[17,43,44] imply that nuclear DNA replication in budding yeast is initiated when two CMG helicases bind to DNA at origins (Fig. 4c). A Pol α-primase complex (red) binds via Ctf4 (not shown) to the leading face of the CMG, relative to the direction of fork motion. On the trailing face of each CMG is a four-subunit Pol ε holoenzyme complex (blue). The catalytic Pol ε N-terminal regions (εN) are not yet bound to DNA in an enzymatically productive manner. The Pol ε C-terminal region is bound to CMG via GINS (not shown). After initial binding to duplex DNA (Fig. 4c), the two CMG complexes transition to bind single-stranded DNA (Fig. 4d) while moving past each other in opposite directions in an ATP-dependent manner. Two ssDNA templates are then available for Pol α to initiate replication on opposite DNA strands. Based on these facts, we suggest two possible models for Pol δ involvement into the initiation of leading-strand replication.

In the first model, Replication Factor C (RFC) first loads Proliferating Cell Nuclear Antigen (PCNA) (not shown) onto the primers made by Pol α (centered red bars in Fig. 4e). Pol δ (green) then binds and rapidly synthesizes nascent-leading strand DNA in a PCNA-dependent manner (Fig. 4e). This synthesis proceeds for about 160 nucleotides (Fig. 4b), until Pol δ approaches Pol ε associated with the other CMG, which has not yet encountered a primer terminus from which to begin synthesis. Pol δ then undergoes "collision release"[17,45,46]. This allows the N-terminal catalytic domain of Pol ε to engage the primed DNA (Fig. 4f) and, using PCNA as a co-factor[17,47], to begin rapid and processive synthesis of leading-strand DNA. Given that the rate of replication is rapid at about 2000 nucleotides per min (ref. [17] and references therein), these processes could all occur within a few seconds. This is the mechanism of initiation suggested by Diffley and colleagues earlier this year (ref. [17] and reviewed in ref. [3]) based on in vitro studies of the minimal replication complex.

In the second model, Pol ε could extend from the initial primer laid down by Pol α (Fig. 4g). This idea is central to many previous studies. The initial primer synthesized by Pol α (labeled * in Fig. 4g, h) would have to be completely excised via extensive nick translation synthesis, presumably performed by the Pol δ that extended from the next Pol α priming event (the first, non-canonical Okazaki fragment; Fig. 4g, h). Note that there is as yet no direct evidence for such extensive nick translation by Pol δ, and given the relative positions of Pol α and Pol ε in replisome structures, it is difficult to see how Pol α associated with one CMG could directly prime synthesis by Pol ε associated with the other CMG. For these reasons, we favor the first model for Pol δ participation in initiation. That said, both models are consistent with the HydEn-seq data (Fig. 4i) and the two models are not mutually exclusive. Both mechanisms could operate depending on yet to be determined circumstances.

The data presented here are consistent with two main ideas. First, in contrasting *pol2-4* and *pol2-16* mutants (Figs. 1 and 2) in two strain backgrounds, it is clear that the compromised phenotypes and ribonucleotide strand biases of the *pol2-16* mutant must primarily result from loss of Pol ε's polymerase activity during leading-strand synthesis, rather than from loss of its proofreading exonuclease. Therefore, given the leading-strand ribonucleotide incorporation bias (Fig. 3b), as well as a preponderance of previous studies, Pol ε is the major DNA polymerase used to replicate the leading strand of the eukaryotic nuclear genome. Second, HydEn-seq end densities indicate that Pol δ contributes to the synthesis of both nascent strands immediately at origins, likely by initiating leading-strand replication.

## Methods

**Yeast strains construction**. *S. cerevisiae* strains used in this study are listed in Supplementary Data 1. All yeast strains (except strains used to analyze genetic interactions of *pol2-16* with *pif1Δ* and *pol32Δ*, see below) were isogenic derivatives of either Δ|(−2)|-7B-YUNI300, representing the Δ7 background, or AC402 and AC403, representing the W303 background. Diploids of Δ7 background were generated from haploid strain using YEpHO as described earlier[14]. Wild-type diploids of W303 background were generated by crossing AC402 and AC403 haploids.

The *pol2-16* mutation was introduced to diploid *S. cerevisiae* via two-step gene replacement, using YIPlac211-pol2-16 plasmid, linearized with BlpI to target integration to the *POL2* locus. Ura+ recombinants were selected on SC-URA and toothpicked twice onto 5-FOA plates to select *pol2-16/POL2* that have excised the plasmid. The presence of the *pol2-16/POL2* alleles was confirmed by PCR (primers: POL2_-391_f, POL2_1102_r, and POL2_695_r; PCR product size: *POL2* – 1509 bp, *pol2-16* – 1102 bp). To exclude the presence of single nucleotides changes in the *pol2-16* locus, the *pol2-16* locus in the heterozygous diploids was amplified as two fragments (N and C). Primers used to amplify fragment N: POL2_-391_f, POL2_695_r, and fragment C: POL2_3423_f and POL2_8156_r. Sequencing

results were analyzed using CLCGenomics Workbench 8.5.1. Primers sequences are listed in Supplementary Data 1.

The yeast bearing pol2-4, pol1L868M, pol2M644G, or pol3L612M polymerases variants were constructed via integration-excision method using plasmids: YIpJB1[26], pYIAL30-pol1L868M[8], p173-pol2M644G[6], and p170-pol3L612M[8], respectively.

Strains with deletion of RNH201 (rnh201Δ) were constructed using one-step gene disruption. PCR product containing hygromycin B—resistance cassette (HPH) and region about 300 nt upstream and downstream of RNH201 was amplified from genomic DNA of SNM106 using primers: RNH35-5′_flank_f and RNH-3′_flank_r. The presence of rnh201Δ::HPH in transformants that were Hygr$^R$ was confirmed by PCR using primers: intHPH-3′_for and RNH35-3′_down1.

Strains with TAP-tag were constructed using POL2-TAP-tag-HIS3MX6 cassette amplified using primers: POL2_6425_f and POL2_6902_r from genomic DNA of YSC1178-202233129 (Dharmacon). The presence of TAP-tag in transformants that were HIS⁺ was confirmed using PCR with primers: POL2_383_f, POL2_2419_f, and conf_5′_TAP_tag.

Yeast strains used to analyze genetic interactions of pol2-16 with pif1Δ and pol32Δ were constructed as follows: the yWH1096 strain was crossed with YAM. pol32Δ::HPH and diploids were sporulated and tetrads were dissected, pol2-16 (QW670) and pol32::HPH (QW671) segregants were selected. Then QW670 and QW671 were crossed to obtain tetrads, from which pol2-16 (QW676) and pol32Δ:: HPH (QW675) were selected. QW675 and QW676 were crossed, yielding a pol2-16 segregant, QW681. Strain QW699 was created by inserting the HPH gene 52 bp after the ORF of POL2 in strain QW681 by standard "ends-out" gene targeting techniques[48] using the HPH (hygromycin B resistance) module in plasmid pAG32[49], amplified by PCR using primers: Oligo_1 and Oligo_2 (Supplementary Data 1) with 50 nt homology to the adjacent regions and 20 nt overlap with the HPH cassette. Strain QW693 resulted from transformation of QW688 with pif1Δ:: KAN cassette, copied from a strain in the yeast knockout collection (GE Healthcare).

**Plasmid construction.** The integration vector bearing pol2-16 allele was constructed as follows: the pol2-16 allele was amplified using genomic DNA from CWY2201 as template and primers: pol2-16_-364_AvaI and pol2-16_4280_SacI with using KAPA HiFi polymerase (Kapa Biosystems). The restriction digestion of PCR product and integration vector YIplac211 with AvaI and SacI was followed with the DNA purification and subsequent ligation and resulted in the integration vector YIplac211-pol2-16 [pol2-16, URA3].

**Doubling time.** Five mililiters of YPDA supplemented with additional adenine (90 μg ml⁻¹) was inoculated with a single yeast colony (wild type and pol2-4) or a spore colony (pol2-16) and incubated at 23 °C with rotation (160 r.p.m.). The doubling times ($D_t$) of yeast strains were calculated from measurements of the OD$_{600}$ of exponentially growing yeast cultures over 3.5-day time course. $D_t$ were calculated using four to six independent biological replicates of each strain, according to the equation: doubling time = $t/g$, where $g = (\log 10 \; (N_t/N_0))/0.3$, $N_0 = $ OD$_{600}$ at start, $N_t = $ OD$_{600}$ at the end, $t = $ time cultured. Presented are the average $D_t \pm$ standard deviation.

**Flow cytometry.** Pre-cultures: 5 ml of YPDA supplemented with additional adenine (90 μg ml⁻¹) were inoculated with a single yeast colony (WT or pol2-4 mutant) or a freshly isolated spore colony of pol2-16 mutant, and incubated at 23 °C with rotation. Pre-cultures were diluted and grown until mid-log phase (OD$_{600}$: 0.3–0.8), then diluted to OD$_{600} = 0.2$, and synchronized with alpha-factor (final concentration in the media 10 μg ml⁻¹) for 4 h (after 2 h, an extra portion of alpha-factor was added to final concentration 10 μg ml⁻¹). Then yeast were collected (3 k.r.p.m., 3 min), washed twice with 25 ml of water and 25 ml of YPDA. After the second wash, yeast cells were resuspended in 24 ml of YPDA supplemented with additional adenine (90 μg ml⁻¹). Yeast samples were collected every 20 min, mixed with sodium azide (final concentration 0.2%), and stored on ice. Yeast cells were collected 3 k.r.p.m., 3 min, RT and fixed in 70% ethanol, then the DNA was stained with propidium iodide according to the standard protocol. Briefly, yeast cells were resuspended in 50 mM Tris-HCl, pH 8.0 with RNase A (final concentration 1 mg ml⁻¹) and incubated for 4 h at 37 °C. Cells were collected (3 k.r.p.m., 3 min) and incubated with pepsin (final concentration 5 mg ml⁻¹) for 1 h at 37 °C and neutralized with 100 mM Tris-HCl pH 8.0. Cells were stained with propidium iodide (final concentration 50 μg ml⁻¹) overnight at 4 °C, then diluted with 50 mM Tris-HCl, pH 8.0 and sonicated. The DNA content was analyzed using the LSR II flow cytometer and FACSDiva software (BD Bioscience). Data were collected for 10 000 cells per sample. Cells were excited using a 488 nm argon laser and emission was detected at 585 nm. Additionally, the cell cycle distribution profiles were analyzed using FlowJo software (FlowJo, LLC). Cells were gated on a PE-area versus PE-width to eliminate doublets.

**Yeast cells staining.** Exponentially growing yeast at 23 °C (OD$_{600}$: ~0.5) in the synthetic complete media supplemented with adenine, was incubated for 1 h with 1 μg ml⁻¹ DAPI. Cells were collected and washed twice with PBS and fixed in 70% ethanol. Before imaging, yeast were collected and washed twice with water. Cells

were imaged using confocal microscope (LSM 710, Carl Zeiss, Inc, oil objective ×63, digital zoom ×2).

**Immunoblotting.** A total of 10 OD units of yeast cells collected at log phase (OD$_{600}$: 0.3–0.8) were resuspended in lysis buffer (150 mM NaCl, 40 mM HEPES, and 1 mM DTT) supplemented with protease inhibitors (complete EDTA-free protease inhibitors, Roche), 2 mM PMSF, and lysed by vortexing with glass beads at 4 °C. TAP-tagged polypeptides were detected using peroxidase-anti-peroxidase antibody (PAP, Sigma, P1291) at 1:2000 dilution. As loading control, an antibody against PSTAIR (Sigma, P7962) was used at 1:5000 dilution. Proteins immobilized on membranes were visualized using chemiluminescent substrates for HRP (WesternBright Sirius, advansta), images were taken using G:BOX (SYNGENE). The resulting bands were quantified using Image Quant TL (GE Healthcare Life Sciences). Band intensities measured for six to seven independent isolates were used to calculate the relative level of Pol2p.

**HydEn-seq libraries construction.** Ribonucleotides were mapped in the genomic DNA using hydrolytic end sequencing technique (HydEn-seq) described by Clausen et al.[11] Briefly, DNA was isolated from exponentially growing yeast (OD$_{600}$: 0.5–1) with using a kit for genomic DNA isolation from yeast (Epicenter). Libraries for Next Generation Sequencing were prepared with using 1 μg of DNA, which was treated with 20 U of restriction enzyme SbfI-HF (New England Biolabs) at 37 °C for 1 h. Next DNA was treated under 300 mM KOH at 55 °C for 2 h and precipitated with ethanol. Library were constructed as described earlier[11]. Briefly, DNA was denatured for 3 min at 85 °C, phosphorylated with 10 U of T4 PNK (#M0236, New England Biolabs) and purified with 1.8 volume of magnetic beads (Magbio). After denaturation for 3 min at 85 °C, DNA fragments were ligated with adapter ARC140 overnight at 25 °C using 10 U of T4 RNA ligase (#M0204, New England Biolabs). This was followed by DNA purification with magnetic beads and the second strand synthesize using 4 U of T7 DNA polymerase (#M0274, New England Biolabs) and ARC76/77 duplex. After DNA purification with magnetic beads, unique indexes were added to DNA fragments using KAPA HiFi HotStart Ready Mix (#KK2602, KAPA Biosystems). The library concentrations and sizes were determined using Bioanalyzer. Libraries were pooled and subject for paired-end sequencing with Illumina HiSeq 2500.

**Alignment and normalization.** HydEn-seq samples were processed as in ref. [11]. Adapter sequence was trimmed from paired-end reads using cutadapt 1.12, discarding pairs where one or both mates were shorter than 15 nt (-m 15, -q 10). Reads derived from oligos used in library preparation were filtered by aligning mate 1 of each pair to an index containing these sequences, and retaining only those that were not mappable (bowtie 1.2, -v2). Retained pairs were subsequently mapped to the S. cerevisiae W303 assembly, retaining unique alignments, and trimming a single nucleotide from the 3′ end of each read to allow alignment of 100% overlapping pairs (bowtie 1.2, -m 1 -v 2 -3 1-best -X1000). Mate 1 of remaining unmapped pairs was then aligned to W303, again retaining only uniquely mappable reads (bowtie 1.2, -m 1 -v 2–best). The positions of the 5′ ends of all uniquely mapped mate 1 reads, from both the paired-end and single-end alignments, were shifted upstream by 1, the implied ribonucleotide location, and bedGraph files containing per-nt counts were generated for each sample using custom scripts.

For the purpose of determining normalization factors, the reads mappable as pairs or mate 1 alone, to multiple locations, were re-aligned with bowtie 1.2 using the same parameters, but omitting -m 1, resulting in a single best alignment for each. BedGraph files were generated as above using all uniquely and non-uniquely mapped reads. From these files, counts of 5′ ends mapping to all SbfI-HF restriction sites, on both strands, were determined. Normalization factors were then calculated using the method implemented in DESeq[50]: for each position, the geometric mean of counts for all samples was determined, as well as the ratio of each sample's count to this value. Sites where zero reads were observed in one or more samples were excluded. The median ratio among all SbfI-HF sites was selected as the normalization factor for each sample. Normalized HydEn-seq counts were subsequently calculated by means of division by these factors.

**Identification of origins of replication in W303.** Origins of replication in the W303 assembly were identified with those annotated to the L03 assembly, previously utilized by Clausen, et al.[11] To this end, 101 nt sequences centered on each L03 ACS were extracted and aligned to W303 with blat v. 34, using default parameters. Of the 214 sequences examined, 208 were mapped to the same chromosome and strand at full length, five sequences mapped to the same chromosome and strand at 95% of full length or better, while a single sequence was mapped at 90%. In a single case, the ACS differed by 1 nt between L03 and W303, while for the remaining 213, the ACS was identical. For each alignment, the ACS location in W303 was determined based on its position relative to the L03 sequence's first mapped base.

**End-count normalization and meta-analysis.** The sum of 5′ ends in bins of 5 bp centered on ACS positions were calculated using BedGraph files and a custom script (heatmap script). These calculations were performed for same and opposite strand independently. Then read counts for 214 origins of replication in each bin

were sum up and scaled (divided) with the normalization factors calculated based on SbfI-HF restriction sites determined in each sample.

**Data availability**. The data discussed in this publication have been deposited in NCBI's Gene Expression Omnibus[51] and are accessible through GEO Series accession number GSE101698. The data that support the findings of this study are available from the corresponding author upon request.

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

## Acknowledgements

We thank K. Bebenek and D. Gordenin for critical reading of and thoughtful comments on the manuscript. We thank all members of DNA Replication Fidelity Group as well as D. Gordenin, P. Mieczkowski, F. Boellmann, and M. Schellenberg for helpful discussions throughout the work. We are thankful to A. Clausen for original instruction on how to prepare HydEn-seq libraries; members of the Fluorescence Microscopy and Imaging Center/NIEHS for help with yeast imaging; members of the Flow Cytometry Center/NIEHS for help with Fluorescence-Activated Cell Sorting (FACS) analysis; members of the High Throughput Genomic Sequencing Facility/UNC Chapel Hill for performing the whole-genome sequencing; C. Wittenberg (Scripps Research Institute) for providing *pol2-16* mutation source and A. Chabes (Umeå University) for providing the AC402 and AC403 strains. This work was supported by Project Z01 ES065070 to T.A.K. from the Division of Intramural Research of the NIH, NIEHS. Work in the Haber lab was supported by NIH grants GM76020 and P01grant GM105473.

## Author contributions

M.A.G., S.A.L. and T.A.K. designed the experiments. M.A.G. and P.B.C. constructed the experimental system. M.A.G. and P.B.C. carried out the in vivo experiments. M.A.G. constructed HydEn-seq libraries. S.A.L., A.B.B. and M.A.G. performed the HydEn-seq data analysis. Z.-X.Z. suggested restriction enzyme usage to normalize the HydEn-seq data. Q.W. and J.E.H. designed and performed the synthetic genetic interactions experiments. T.A.K., M.A.G. and S.A.L. wrote the manuscript, and all authors edited the manuscript.

## Additional information

**Competing interests:** The authors declare no competing interests.

