## [Peer Review File · Nature Communications]

Reviewers' comments:

Reviewer #1 (Remarks to the Author):

Building on their past work, the authors probe deeper into the division of labour between Pol epsilon and Pol delta at budding yeast replication forks. The manuscript presents three main conclusions:

(1) replication and cell growth are both very slow in pol2-16 cells that lack the catalytic domain of Pol epsilon. Previous work showed that pol2-16 was viable, but the present study emphasises just how sick this mutant actually is, before the acquisition of suppressor mutations that aid growth.

(2) Pol delta synthesises both leading and lagging strand in pol2-16 cells (this was the predicted result but was never tested previously).

(3) During initiation at origins, both strands seem to be dependent upon Pol delta, before a switch to Pol epsilon for leading strand synthesis after a couple of hundred nucleotides. The authors present two models – either Pol delta initially synthesises the leading strand before a switch to Pol epsilon (matching suggestions based on a reconstituted budding yeast replisome, and also from in vivo work with fission yeast), or else Pol epsilon does indeed synthesise the leading strand from the beginning, but this fact is masked at origins by extensive nick-translation synthesis by Pol delta.

The issues addressed in the manuscript are important and the data are of high quality. It could be argued that the data don't contain very many surprises, and essentially just reinforce the current view. But this would probably be a little harsh, given the controversy regarding the role of Pol epsilon and Pol delta in leading strand synthesis, and these findings will be of considerable interest to those in the DNA replication field. I would support publication in Nature Communications, subject to the authors dealing with a couple of minor points.

Specific points:

1. The first section emphasises that cells lacking the Pol epsilon catalytic domain are extremely sick and replication is aberrant, despite the fact that 'viable' colonies can be isolated. However, it would be helpful if the authors could compare the level of the Pol2 protein in wild type and pol2-16 cells, since a trivial reason for the sickness/near-inviability of Pol2-16 could be that the truncated protein is very unstable, leading to initiation defects, in addition to leading-strand replication by Pol delta. This possibility might be relevant to the easy generation of suppressor mutations that greatly improve the growth of pol2-16, since these might involve mutations in pathways that degrade poorly folded proteins. The authors could compare the level of Pol2 in

slow-growing colonies of pol2-16 (without one of the presumed suppressors) and in fast-growing colonies of pol2-16 (with suppressor).

2. The authors use the data in Supplementary Table 3 to propose that “A suppressor of pol2-16 growth defect is apparently heterozygous in the BY4733 x QW699 diploid”. They could prove this by isolating POL2 colonies from this diploid, 50% of which should have the suppressor. The presence of the suppressor could be tested by crossing such POL2 colonies once again to pol2-16. In this way, it should subsequently be possible to set up a cross of pol2-16+suppressor to POL2+suppressor, from which cross all the pol2-16 spores should give rise to fast growing colonies.

Reviewer #2 (Remarks to the Author):

There has been a controversy regarding the identity of the DNA polymerase responsible for synthesis of the leading strand in eukaryotes. A wealth of genetic, genomic and biochemical evidence points to a major role for pol epsilon, but a paper in 2015 argued that pol delta is the major leading strand polymerase, with epsilon primarily acting to edit mistakes from delta. In this paper the authors use a variety of approaches to re-examine this issue. They show that the pol2-16 mutant, which lacks the catalytic domain of pol epsilon, is extremely slow-growing after sporulation, but appears to ‘adapt’ with time – interpreted by the authors as accumulation of extragenic suppressors (more below). They show these properties are not shared by the exonuclease-deficient pol2-14 mutant, arguing that the polymerase, not the exonuclease, is critical for growth. Using a whole genome approach to measure incorporation of ribonucleotides in strains with mutant polymerases which incorporate ribonucleotides at elevated rates, they provide further evidence that epsilon synthesizes the leading strand, and delta and alpha synthesizes the lagging strand, even in a different strain background (W303) from the one used in their original analysis. They also show that delta synthesizes the leading strand in the pol2-16 mutant. By a novel, detailed analysis of their data, they also show that delta synthesises the first ~1-200 nucleotides of the leading strand, consistent with previous genetic and biochemical experiments. Overall this is an excellent paper; the experiments have been carefully planned and well-executed. I have only one quibble. The authors infer that the differences in growth rates of different isolates of pol2-16 are due to accumulation of extragenic suppressors. This is certainly a possibility, but the evidence presented is quite weak. They should either present proper spore segregation data in a clean background or entertain the possibility that epigenetics or non-encoded metabolic adaptations may also contribute to differences in growth rates. Otherwise, I think this paper makes an important contribution to the field and I fully support publication.

Reviewer #3 (Remarks to the Author):

This submission addresses the division of labor among DNA polymerases during eukaryotic DNA replication, an important issue subject to intense study in the last decade. Analyses of strand-specific mutations and rNTP incorporation analyses in yeast suggest that within the eukaryotic replication fork, polymerase epsilon catalyzes the synthesis of the leading strand whereas polymerase delta catalyzes the synthesis of the lagging strand. This polymerase assignment was in agreement with biochemical data in reconstituted replication forks, but appeared to contradict other observations, including a recent study suggesting that polymerase delta is primarily responsible for synthesizing both strands and that the different mutagenesis patterns reflect differential, strand-specific repair and proofreading activities.

The current submitted paper reports growth and rNTP incorporation patterns in previously analyzed mutants of polymerase epsilon lacking catalytic and exonuclease domains. Some experiments were repeated in two different strain backgrounds to allay concerns about the effects of strain backgrounds on the previously published data. These data support the primary role of polymerase epsilon in leading strand synthesis. The manuscript also show data implying that polymerase delta can catalyze leading strand synthesis, albeit with a requirement for co-factors, in the absence of polymerase epsilon. The data also imply that polymerase delta is involved in the early stages of leading strand synthesis.

The results reported in the paper suggest that when pol epsilon is missing, pol delta can support growth while exhibiting a requirement for pif1 and the pol32 subunit. During normal replication, the patterns of rNTP incorporation suggest that pol delta starts replication at origins both for the leading and the lagging strands, presumably handing off replication after synthesizing a little more than one Okazaki fragment length. The need for co-factors is interesting and the paper will benefit from further discussion and clarification of this point.

The data reported in the paper convincingly support the notion that the polymerase and not the exonuclease of polymerase epsilon is responsible for the replication-related phenotypes,. However, the observation that the 2-4 mutant exhibits small colonies in 23C in one of the backgrounds but not the other should be discussed.

Minor

Line 33 parenthesis after Ref. 4

Figure 2c should be mentioned in the main text after 2b and before 3a

Format reference Kesti et al line 199

Line 290 parenthesis after ref. 3

Responses to Reviewers' Comments:

Reviewer #1

Comments:

Building on their past work, the authors probe deeper into the division of labour between Pol epsilon and Pol delta at budding yeast replication forks. The manuscript presents three main conclusions:

(1) replication and cell growth are both very slow in *pol2-16* cells that lack the catalytic domain of Pol epsilon. Previous work showed that *pol2-16* was viable, but the present study emphasises just how sick this mutant actually is, before the acquisition of suppressor mutations that aid growth.

(2) Pol delta synthesises both leading and lagging strand in *pol2-16* cells (this was the predicted result but was never tested previously).

(3) During initiation at origins, both strands seem to be dependent upon Pol delta, before a switch to Pol epsilon for leading strand synthesis after a couple of hundred nucleotides. The authors present two models – either Pol delta initially synthesises the leading strand before a switch to Pol epsilon (matching suggestions based on a reconstituted budding yeast replisome, and also from *in vivo* work with fission yeast), or else Pol epsilon does indeed synthesise the leading strand from the beginning, but this fact is masked at origins by extensive nick-translation synthesis by Pol delta.

The issues addressed in the manuscript are important and the data are of high quality. It could be argued that the data don't contain very many surprises, and essentially just reinforce the current view. But this would probably be a little harsh, given the controversy regarding the role of Pol epsilon and Pol delta in leading strand synthesis, and these findings will be of considerable interest to those in the DNA replication field. I would support publication in Nature Communications, subject to the authors dealing with a couple of minor points.

Specific points:

Comment 1. The first section emphasizes that cells lacking the Pol epsilon catalytic domain are extremely sick and replication is aberrant, despite the fact that 'viable' colonies can be isolated. However, it would be helpful if the authors could compare the level of the Pol2 protein in wild type and *pol2-16* cells, since a trivial reason for the sickness/near-inviability of *Pol2-16* could be that the truncated protein is very unstable, leading to initiation defects, in addition to leading-strand replication by Pol delta. This possibility might be relevant to the easy generation of suppressor mutations that greatly improve the growth of *pol2-16*, since these might involve mutations in pathways that degrade poorly folded proteins. The authors could compare the level of Pol2 in slow-growing colonies of *pol2-16* (without one of the presumed suppressors) and in fast-growing colonies of *pol2-16* (with suppressor).

Response:

We have taken the reviewer's suggestion and measured the level of Pol2p in wild type cells and in freshly isolated *po2-16* cells by Western Blot analysis. The results indicated that there is indeed a higher level of degradation of Pol2p in *pol2-16* yeast as compared to wild type cells. The level of non-degraded Pol2p in the *pol2-16* freshly isolated spore-colonies is about 30% of

the level observed in the wild type cells. This suggests that the truncated Pol2p in *pol2-16* is unstable and may indicate that the N-terminal portion of Pol2p (bearing the catalytic domains) is critical to stabilize the C-terminal part of Pol2p. It has been shown that the C-terminal part of Pol2p interacts with Dpb2p, and that this bridges the interaction of Pol ϵ with the GINS complex. Both Pol ϵ as well as GINS are components of CMGE helicase which formation is essential for initiation of chromosomal DNA replication^{1,2}. Because of the lower level of Pol2p in the *pol2-16* mutants, the initiation of DNA replication may be impaired, which could be one of the reasons for the extremely slow growth of the *pol2-16* mutants.

We added this result to the result section (page 4, lines 102-105 and Figure 1f-g) as well as a short discussion (page 8-9, lines 220-227).

In the future, we plan to compare the level of Pol2p in freshly isolated *pol2-16* with the level in faster-growing *pol2-16* (bearing the suppressor/-s). Understanding the mechanism/-s of suppression of lack of Pol2 catalytic domains will be subject of a subsequent publication.

Figure 1

Comment 2. The authors use the data in Supplementary Table 3 to propose that “A suppressor of *pol2-16* growth defect is apparently heterozygous in the BY4733 x QW699 diploid”. They could prove this by isolating *POL2* colonies from this diploid, 50% of which should have the suppressor. The presence of the suppressor could be tested by crossing such *POL2* colonies once again to *pol2-16*. In this way, it should subsequently be possible to set up a cross of *pol2-16*+suppressor to *POL2*+ suppressor, from which cross all the *pol2-16* spores should give rise to fast growing colonies.

Response

We have carried out the experiment suggested by the reviewer. The results support the suggestion that in the particular yeast strain that we used (QW699 and isogenic QW710), there is an unlinked suppressor mutation that influences the growth of *pol2-16*. We crossed a QW710, a NAT-marked *pol2-16* strain, isogenic to QW699, and which presumably carries a suppressor, to the *POL2* strain, BY4733 and obtained 14 *MATa POL2* segregants that could be backcrossed to the QW710, which is *MATα*. The resulting diploids were then sporulated and tetrads were dissected. The results of 4 such dissections are shown in Supplementary Fig. 4. In 7 of the backcrosses, the proportion of *pol2-16* segregants was approximately 50% of *POL2* segregants (Supplementary Fig. 4, examples A and B and Supplementary Table 4), whereas at least 5 of the remaining 7 backcrosses viability of the NAT-marked *pol2-16* was comparable or nearly (>85%) comparable (Supplementary Fig. 4, examples A and B and Supplementary Table 4). Two results were intermediate. The variation in the size of the colonies and the residual poor growth of some *pol2-16* segregants could still be attributed to other modifiers. Based on above analysis, we cannot exclude that there are other mechanisms of suppression of lack of Pol2p catalytic domains, such as multiple mutations that improve fitness or epigenetic changes or non-encoded metabolic adaptations. These possibilities are now stated in the Discussion.

Reviewer #2

Comments

There has been a controversy regarding the identity of the DNA polymerase responsible for synthesis of the leading strand in eukaryotes. A wealth of genetic, genomic and biochemical evidence points to a major role for pol epsilon, but a paper in 2015 argued that pol delta is the major leading strand polymerase, with epsilon primarily acting to edit mistakes from delta. In this paper the authors use a variety of approaches to re-examine this issue. They show that the *pol2-16* mutant, which lacks the catalytic domain of pol epsilon, is extremely slow-growing after sporulation, but appears to 'adapt' with time – interpreted by the authors as accumulation of extragenic suppressors (more below). They show these properties are not shared by the exonuclease-deficient *pol2-4* mutant, arguing that the polymerase, not the exonuclease, is critical for growth. Using a whole genome approach to measure incorporation of ribonucleotides in strains with mutant polymerases which incorporate ribonucleotides at elevated rates, they provide further evidence that epsilon synthesizes the leading strand, and delta and alpha synthesizes the lagging strand, even in a different strain background (W303) from the one used in their original analysis. They also show that delta synthesizes the leading strand in the *pol2-16* mutant. By a novel, detailed analysis of their data, they also show that delta synthesises the first ~1-200 nucleotides of the leading strand, consistent with previous genetic and biochemical experiments. Overall this is an excellent paper; the experiments have been carefully planned and well-executed. I have only one quibble. The authors infer that the differences in growth rates of different isolates of *pol2-16* are due to accumulation of extragenic suppressors. This is certainly a possibility, but the evidence presented is quite weak. They should either present proper spore segregation data in a clean background or entertain the possibility that epigenetics or non-encoded metabolic

adaptations may also contribute to differences in growth rates. Otherwise, I think this paper makes an important contribution to the field and I fully support publication.

Response

We cannot yet make any conclusions about the nature of the suppression that is occurring in *pol2-16* cells. The reviewer describes two possibilities, epigenetics or non-encoded metabolic adaptations. We are pleased to add these possibilities to the discussion (page 8, lines 217 - 219).

Reviewer #3

Comments

This submission addresses the division of labor among DNA polymerases during eukaryotic DNA replication, an important issue subject to intense study in the last decade. Analyses of strand-specific mutations and rNTP incorporation analyses in yeast suggest that within the eukaryotic replication fork, polymerase epsilon catalyzes the synthesis of the leading strand whereas polymerase delta catalyzes the synthesis of the lagging strand. This polymerase assignment was in agreement with biochemical data in reconstituted replication forks, but appeared to contradict other observations, including a recent study suggesting that polymerase delta is primarily responsible for synthesizing both strands and that the different mutagenesis patterns reflect differential, strand-specific repair and proofreading activities.

The current submitted paper reports growth and rNTP incorporation patterns in previously analyzed mutants of polymerase epsilon lacking catalytic and exonuclease domains. Some experiments were repeated in two different strain backgrounds to allay concerns about the effects of strain backgrounds on the previously published data. These data support the primary role of polymerase epsilon in leading strand synthesis. The manuscript also show data implying that polymerase delta can catalyze leading strand synthesis, albeit with a requirement for co-factors, in the absence of polymerase epsilon. The data also imply that polymerase delta is involved in the early stages of leading strand synthesis.

The results reported in the paper suggest that when pol epsilon is missing, pol delta can support growth while exhibiting a requirement for pif1 and the pol32 subunit. During normal replication, the patterns of rNTP incorporation suggest that pol delta starts replication at origins both for the leading and the lagging strands, presumably handing off replication after synthesizing a little more than one Okazaki fragment length. The need for co-factors is interesting and the paper will benefit from further discussion and clarification of this point.

Response

We have added the following brief discussion on the need for co-factors to the Discussion, page 10, lines 264-268.

“It is possible that the requirement for *POL32* and *PIF1* when Pol ϵ catalytic activity is absent is similar to their requirement in extensive DNA synthesis during break-induced replication, where

leading and lagging-strand DNA synthesis are not coupled³ and where the initial DNA synthesis appears to be dependent on Pol δ , with Pol ϵ only being required at a later stage^{4,5}.”

Comment

‘The data reported in the paper convincingly support the notion that the polymerase and not the exonuclease of polymerase epsilon is responsible for the replication-related phenotypes. However, the observation that the 2-4 mutant exhibits small colonies in 23C in one of the backgrounds but not the other should be discussed.’

Response

The difference in the spore-colony sizes of the *pol2-4/POL2* yeast between the $\Delta 7$ and W303 backgrounds is related to the difference in the doubling time between the $\Delta 7$ and W303 yeast backgrounds, and may be due to one or more of over 10 thousand SNPs detected by the whole genome sequencing (data not shown, available upon request). This is now briefly discussed in the legend for Figure 1, page 29, lines 795 - 797.

Minor Comments

Line 33 parenthesis after Ref. 4

Figure 2c should be mentioned in the main text after 2b and before 3a

Format reference Kesti et al line 199

Line 290 parenthesis after ref. 3

Response

Each of these minor corrections has been made.

REFERENCES

1. Sengupta, S., van Deursen, F., de Piccoli, G. & Labib, K. Dpb2 integrates the leading-strand DNA polymerase into the eukaryotic replisome. *Curr Biol* **23**, 543-52 (2013).
2. Muramatsu, S., Hirai, K., Tak, Y.S., Kamimura, Y. & Araki, H. CDK-dependent complex formation between replication proteins Dpb11, Sld2, Pol (epsilon), and GINS in budding yeast. *Genes Dev* **24**, 602-12 (2010).
3. Saini, N. et al. Migrating bubble during break-induced replication drives conservative DNA synthesis. *Nature* **502**, 389-92 (2013).
4. Lydeard, J.R., Jain, S., Yamaguchi, M. & Haber, J.E. Break-induced replication and telomerase-independent telomere maintenance require Pol32. *Nature* **448**, 820-3 (2007).
5. Lydeard, J.R. et al. Break-induced replication requires all essential DNA replication factors except those specific for pre-RC assembly. *Genes & Development* **24**, 1133-1144 (2010).

Reviewers' comments:

Reviewer #1 (Remarks to the Author):

The authors have done a good job of addressing the points that I raised, and I think that the paper is ready to be published.